# Assessment of Enterovirus Excretion and Identification of VDPVs in Patients with Primary Immunodeficiency in India: Outcome of ICMR–WHO Collaborative Study Phase-I

**DOI:** 10.3390/vaccines11071211

**Published:** 2023-07-06

**Authors:** Madhu Chhanda Mohanty, Mukesh Desai, Ahmad Mohammad, Amita Aggarwal, Geeta Govindaraj, Sagar Bhattad, Harsha Prasada Lashkari, Liza Rajasekhar, Harish Verma, Arun Kumar, Unnati Sawant, Swapnil Yashwant Varose, Prasad Taur, Reetika Malik Yadav, Manogat Tatkare, Mevis Fernandes, Umair Bargir, Sanjukta Majumdar, Athulya Edavazhippurath, Jyoti Rangarajan, Ramesh Manthri, Manisha Ranjan Madkaikar

**Affiliations:** 1Mumbai Unit, ICMR-National Institute of Virology (ICMR-NIV), Mumbai 400012, India; 2Department of Immunology, Bai Jerbai Wadia Hospital for Children, Mumbai 400012, India; 3World Health Organization, Country Office, New Delhi 110011, India; 4Department of Clinical Immunology & Rheumatology, Sanjay Gandhi Postgraduate Institute of Medical Sciences, Lucknow 226014, India; 5Department of Pediatrics, Government Medical College, Kozhikode 673008, India; 6Department of Pediatrics, Aster CMI Hospital, Bangalore 560092, India; 7Department of Pediatrics, Kasturba Medical College, Mangalore 576104, India; 8Department of Clinical Immunology and Rheumatology, Nizam’s Institute of Medical Sciences, Hyderabad 500082, India; 9World Health Organization, CH-1211 Geneva, Switzerland; 10ICMR-National Institute of Immunohaematology (ICMR-NIIH), Mumbai 400012, India; 11Multidisciplinary Research Unit, Government Medical College, Kozhikode 673008, India

**Keywords:** primary immunodeficiency disorder (PID), inborn errors of immunity (IEI), Oral Polio Vaccine (OPV), immunodeficiency-related vaccine-derived polioviruses (iVDPV), Global Polio Eradication Initiative (GPEI), enteroviruses, prolonged excretion, polio eradication, iVDPV surveillance, AFP surveillance, combined immunodeficiency (CID)

## Abstract

The emergence of vaccine-derived polioviruses (VDPVs) in patients with Primary Immunodeficiency (PID) is a threat to the polio-eradication program. In a first of its kind pilot study for successful screening and identification of VDPV excretion among patients with PID in India, enteroviruses were assessed in stool specimens of 154 PID patients across India in a period of two years. A total of 21.42% of patients were tested positive for enteroviruses, 2.59% tested positive for polioviruses (PV), whereas 18.83% of patients were positive for non-polio enteroviruses (NPEV). A male child of 3 years and 6 months of age diagnosed with Hyper IgM syndrome was detected positive for type1 VDPV (iVDPV1) with 1.6% nucleotide divergence from the parent Sabin strain. E21 (19.4%), E14 (9%), E11 (9%), E16 (7.5%), and CVA2 (7.5%) were the five most frequently observed NPEV types in PID patients. Patients with combined immunodeficiency were at a higher risk for enterovirus infection as compared to antibody deficiency. The high susceptibility of PID patients to enterovirus infection emphasizes the need for enhanced surveillance of these patients until the use of OPV is stopped. The expansion of PID surveillance and integration with a national program will facilitate early detection and follow-up of iVDPV excretion to mitigate the risk for iVDPV spread.

## 1. Introduction

With the wild poliovirus eliminated from most of the world apart from Pakistan and Afghanistan [1] there is a threat to the Global Polio Eradication program from the reintroduction of the poliovirus and emergence of circulating vaccine-derived poliovirus (cVDPV) [2] in the polio-free countries. Prolonged uninterrupted circulation of vaccine viruses may mutate over the course of infection and reacquire neurovirulence. Primary immunodeficiency (PID) (renamed as inborn errors of immunity (IEI)) cases provide a suitable milieu for the emergence of immunodeficiency-associated vaccine-derived poliovirus (iVDPV) [3]. Over 150 iVDPV cases [4] have been reported worldwide. iVDPVs are strains of the weakened poliovirus contained in OPV which can revert to the neurovirulent form in the immunodeficient patient and cause outbreaks in areas where vaccine coverage is low. 

Primary immunodeficiency diseases are a heterogeneous group of over 450 genetic disorders of the immune system [5], among whom patients with combined immunodeficiency diseases and antibody deficiencies are particularly at a high risk of prolonged excretion of VDPV after vaccination with OPV or close contact with a vaccinated individual [6,7,8,9]. PID patients may continue to excrete VDPV for a long duration asymptomatically and, thereby, act as a source of poliovirus reintroduction in the community. Surveillance for poliovirus circulation is carried out routinely through a robust system of Acute Flaccid Paralysis (AFP) surveillance as well as environmental surveillance through the Global Polio Laboratory Network (GPLN) [10]. In many countries, VDPVs have been reported in AFP surveillance particularly in paralyzed children, who then may receive a diagnosis of one of the PIDs. Hence, expansion of surveillance in PID patients can facilitate early detection and follow-up of iVDPV excretion among patients with PID which will help to mitigate the risk for iVDPV spread and maintain the eradication of all polioviruses.

In India, there have been advances in the diagnosis and management of PIDs under the leadership of the Indian Council of Medical Research (ICMR) as well as the Foundation of Primary Immunodeficiency Diseases (FPID) over the last decade. A systematic research study was conducted as a collaborative effort by the ICMR-National Institute of Virology, Mumbai Unit and ICMR-National Institute of Immunohaematology, World Health Organization, and six FPID centers. In this study, all cases diagnosed as PIDs were screened for the excretion of poliovirus/non-polio enteroviruses.

## 2. Materials and Methods

The diagnosed PID patients who routinely visited the seven hospitals/study sites, namely BJ Wadia Hospital for Children, Mumbai; ICMR-NIIH, Mumbai; SGPIMS, Lucknow; GMC, Kozhikode; ASTER CMI, Bangalore; KMC, Mangalore; and NIMS, Hyderabad, for follow-up and Intravenous Immunoglobulin (IVIg) prophylaxis and the individuals who were newly diagnosed with PID were enrolled in the research study conducted during December 2019 to December 2021. During the lockdown observed for the COVID-19 pandemic in mid-2020, the samples were collected from the patients’ homes by the National Polio Surveillance Program, WHO by obtaining verbal consent from the patient/parents. The research study enrolled 157 diagnosed PID cases of mostly pediatric age with different categories of PIDs such as humoral immunodeficiency (agammaglobulinemic), Common Variable Immunodeficiency Disorder (CVID), Hypogammaglobulinemia, severe combined immunodeficiency (SCID), etc. 

### 2.1. Stool Sample Collection, Transportation and Storage

Stool samples were collected in a clean plastic screw-capcontainer after due consent and stored at −20 °C at the respective centers. Patients whose samples were found positive for enteroviruses in monthly stool samples were followed up until 2 consecutive samples became negative for enteroviruses. In the case of patients who were negative for enteroviruses stool samples were collected at 6-monthly intervals. The stored stool samples collected in a month (5–10 samples approx.) were sent to NIVMU by courier in dry ice. At NIVMU the stool samples were stored at −20 °C for further processing.

### 2.2. Processing of Samples and Enterovirus Isolation

The isolation of enteroviruses from fecal extracts was performed as described in the WHO laboratory manual [11]. Enterovirus culture was performed by using Human Rhabdomyosarcoma (RD) cell-line and a transgenic mouse cellline, expressing poliovirus receptor (L20B). Before inoculation in cell cultures, the fecal extracts (10%, wt/vol) were prepared in phosphate-buffered saline (Sigma cat. no. D8662; Sigma-Aldrich, St. Louis, MO, USA) with 10% (vol/vol) chloroform. A total of 200 μL of stool extracts were inoculated in the cells in duplicates, and incubated at 36 °C, and observed microscopically for cytopathogenic effect (CPE) daily for 5 days. CPE positive cell cultures were freeze-thawed and the culture medium was used for virus identification. The samples were scored as negative if 2 consecutive passages in the same cell line did not produce CPE.

### 2.3. Intratypic Differentiation and Molecular Typing

Intratypic differentiation of cell culture positive samples was performed by real-time PCR (Applied Biosystems, 7500). The frozen CPE-positive cells were thawed and the viral RNA was extracted using a QIAamp Viral RNA Mini Kit (QIAGEN, Hilden, Germany) according to the manufacturer’s instructions. Primer pairs 222/224 and 88/89 were used for partial VP1 sequencing and identification of enterovirus types for NPEVs. In the case of poliovirus isolates, VP1 region amplification (≈900 nt) and reverse transcription PCR was performed in a single tube using reverse primer Q8 and forward primer Y7 [9]. A Big Dye Terminator v3.1 Cycle Sequencing Kit (Applied Biosystems, Foster City, CA, USA) was used for sequencing according to the manufacturer’s instructions. The sequences were resolved on an ABI 3130xl Genetic Analyzer (Applied Biosystems) and edited using Sequencher version 4.10.1 software (Gene Codes, Ann Arbor, MI, USA). The VP1 sequences of poliovirus reference strains obtained from the GenBank database were used for the comparative analysis of nucleotide and amino acid sequences. The program CLUSTAL W (http://www.ebi.ac.uk, accessed on 20 January 2023) embedded in MEGA7 (http://www.megasoftware.net, accessed on 20 January 2023, was used for the alignment of the VP1 region. Further nucleotide and amino acid sequences were analyzed and compared pair wise.

The NPEV sequences were subjected to a BLAST search (https://www.ncbi.nlm.nih.gov/BLAST, accessed on 20 January 2023) and the criteria of >75% nucleotide and >85% amino acid similarity in the VP1 region was followed to define virus type.

### 2.4. Micro-Neutralization Assay for Poliovirus Neutralizing Antibody Titre

Micro-neutralization assay was used to estimate the poliovirus neutralizing antibody levels of the patient [12]. HEp-2 Cincinnati cells were used as the cell substrate and the Sabin poliovirus vaccine strains obtained from NIBSC, UK, were used as the challenge virus. Serial two-fold diluted serum samples (1:8–1:1024) were used for the test. 

### 2.5. Multiplex Cytokine Assay

Pro-inflammatory cytokines (IL-1β, IL-6, IL-8, and TNF-α), Th1 (IFN-α, IFNγ, and TNFα), Th2 cytokines (IL-10, IL-4, and IL-13), and associated cytokines and chemokines (MCP-1, MIP-1α, IP-10, and RANTES) were analyzed using Multiplex Cytokine Analysis Kits (Merck, Milliplex). The assays were run in duplicates according to the manufacturer’s protocol. The results were analyzed in serum samples using the Luminex-100 system Version 1.7 (Luminex, Austin, TX, USA). The MasterPlex QT 1.0 system (MiraiBio, Alameda, CA, USA) was used for data analysis. Sample concentrations were calculated using a five-parameter regression formula from the standard curves. 

### 2.6. Statistical Analysis

Mean cytokine values at different time points of collection were compared using the Student’s *t*-test through Sigma Plot software (Version 10.0). *p* < 0.05 was considered significant. 

## 3. Results

A total of 157 patients with a confirmed PID diagnosis were enrolled in the study during the period of December 2019 to December 2021. Of these, 154 cases could be tested for the presence of enteroviruses as 3 patients passed away before even the collection of first sample. Out of total 154 cases tested, 21.42% were positive for enteroviruses, 4 cases (2.59%) tested positive for poliovirus types (PV), whereas 29 cases (18.83%) were found positive for non-polio enteroviruses (NPEV) at different time points of sample collection. During the course of study, a total of 17 enrolled PID cases did not survive and, therefore, the follow-up sample collection could not be completed for those cases (Table 1). Including all follow-ups, a total of 535 stool samples were collected from the 7 study institutes across India.

Out of four poliovirus positive cases, one case each of iVDPV type1, P3SL, and P1SL, were enrolled at site B.J. Wadia Hospital for Children, Mumbai, Maharashtra, whereas one case of P3SL was enrolled at site KMC, Mangalore, Karnataka. On the other hand, NPEVs were detected in cases enrolled at all seven study sites. Varied types of enterovirus B species, echoviruses, and coxsackieviruses were detected during the course of the Phase-I study (Table 2). Out of 154 PID cases enrolled in the study 71% were males while 29% cases were females. The highest number of PID cases positive for enteroviruses belonged to the age group of 1 to 5 years of age, followed by 5 to 10 years. However, all the poliovirus cases that were identified in the study belonged only to the age group of 1 to 5 years (Figure 1).

### 3.1. Poliovirus Detection

Out of 535 stool samples tested, 5 samples (collected from 4 cases) were confirmed as positive for PVs which includes poliovirus type 1 and type 3. A 3-year-old male child with Hyper IgM syndrome (with a defect in CD40) showed more than 10 nucleotide changes and was, therefore, confirmed as the first vaccine-derived poliovirus (iVDPV) type 1 case of the study. The child had received its last bOPV dose approximately a year ago from the date detected positive for poliovirus. The VP1 region showed 16 nucleotide changes. The collection of consecutive negative samples was delayed by more than 30 days (as per protocol) due to the COVID-19 pandemic-related restrictions and lockdown. The sample collected on 80th day turned out to be negative for polioviruses; therefore, the possibility of missed VDPV samples cannot be ruled out. The patient was followed up routinely throughout the study. The reciprocal anti-polio antibody titer as tested using micro-neutralization assay was found to be 35.92 and 22.63 for type 1 and 7.13 and 11.31 for type 3 polioviruses during pre-post-iVDPV detection, respectively. Comparative analysis of cytokine release by multiplex cytokine assay revealed a significantly high IL-8 during the VDPV clearance period as compared to the sample collected during the diagnosis of PID. Apart from that, IP-10, IL-6, and IFN-alpha titers during VDPV clearance were observed to be significantly decreased as compared to the un-infected period. Another one-month-old female child with SCID excreted PISL with 4 nucleotide changes in the sequences; however, the child passed away before the collection of the second sample. The child had received a birth dose of polio as per the schedule before the confirmation of PID. Similarly, a 5-month-old male child, having received a birth dose of bOPV, excreted P3SL with 3 nucleotide changes in the first isolate, and tested negative for the next consecutive collections. The child passed away during the course of study. The fourth case of a 4-year-old male child with XLA had accidentally taken bOPV during a pulse polio campaign. The sample showed a mixture of two PVs (PV1 + PV3) in the first sample while the consecutive follow-up sample showed only P3SL, which further turned negative in the subsequent follow-ups (Table 3). 

The molecular characterization of iVDPV type 1 isolated from a 3-year-old male child showed five non-synonymous mutations in the VP1 region. Out of those, mutation at nt296 led to amino acid (position 99) reversion from Lys to Thr which is a wild-type substitute. Non-synonymous mutation at nt316 amino acid position 106 showed a substitution of amino acid Ser different from Sabin Thr and Mahoney Ala. The other three non-synonymous changes are not at the attenuated position and different from the Sabin and wild type. A one-month-old SCID child with Sabin-like poliovirus type 1 with four nucleotide changes showed one non-synonymous mutation at the nt299 amino acid position 100 from Asn to Ser which is not seen at the attenuated site and is different from the Sabin and wild type. A 5-month-old male child with WAS was detected with Sabin-like poliovirus type 3 with three nucleotide changes showed one non-synonymous change. The amino acid position 54 (nt161) changed from Ala to Val which takes part in the reversion from the Sabin to the wild type (Table 3). 

### 3.2. Non Polio Enterovirus Detection

Out of 535 stool samples tested, 67 samples, collected from 29 cases, which showed CPE on the RD cell line and remained negative in the L20B cell line were confirmed as positive for NPEVs. CVID was seen to be the most common PID type among the patients testing positive for NPEVs, followed by WAS and Hyper IgM syndrome. The maximum period of NPEV excretion was observed in an adult patient with CVID that continued excreting Echovirus 21 for about 507 days and had been followed continuously. A total of 72.41% of patients excreting NPEVs for varied time durations discontinued excretion at certain time points, whereas about 20% of patients continued excretion until the last sample collection. Two patients expired while their samples tested positive for NPEVs.

Overall, 23 different NPEV types belonging to three different species were detected from 29 cases. The most prominent among all was the EV-B species with 23 (79%) isolates distributed among 17 NPEV types. A total of 17.24% of isolates belonged to the EV-A species, whereas only one isolate belonged to the EV-C species (Table 4). The five most frequently observed NPEV types were E21 (19.4%), E14 (9%), E11 (9%), E16 (7.5%), and CVA2 (7.5%) (Figure 2).

At the end of the Phase-I study, none of the PID patients showed excretion of poliovirus, whereas only four patients were excreting NPEVs. A total of 2 patients excreting poliovirus were alive but had stopped excreting the virus, whereas 18 alive patients had stopped excreting NPEVs. Two patients excreting poliovirus and two excreting NPEVs passed away during the course of the study. For patients excreting NPEVs, 5 patients could not be followed up due to varied reasons (Table 5).

### 3.3. Enterovirus Excretion among IVIg Recipients and Non-Recipients

A total of 97 PID patients with antibody deficiency and combined immunodeficiency required IVIg prophylaxis. However, 18.5% of patients among them didn’t receive IVIg. A total of 68.96% of IVIg receivers tested positive for NPEVs. These patients include 18 NPEV-positive and 3 PV-positive patients. No significant difference in EV excretion could be observed between the IVIg receivers and non-receivers group (Figure 3).

### 3.4. Immunodeficiency Category Wise Distribution of PID Cases

Although the total number of patients enrolled with Primary Antibody Deficiency (PAD) was higher, enterovirus excretion was found significantly more in patients with Combined Immunodeficiency (CID), than other immunodeficiencies (*p* < 0.05) (Figure 4). SCID showed the highest detection rate (31%) among all the CID categories with a higher number of enterovirus infections as compared to others (Appendix A).

## 4. Discussion

The long-term excretion of iVDPVs by PID patients can jeopardize polio eradication efforts. Therefore, GPEI has proposed integrating poliovirus surveillance in PID patients with AFP and environmental surveillance and supporting the implementation of iVDPV surveillance in several countries in order to identify non-paralyzed iVDPV patients [13]. Multiple campaigns with a monovalent OPV type 2 vaccine have been conducted due to the persistent transmission of cVDPV2 to control outbreaks [14]. It is an important strategy to maintain high vaccination coverage in the community while preventing cVDPV. An inactivated polio vaccine (IPV) has been introduced in all countries as of May 2019 to achieve and sustain a polio-free world and as part of the preparations for the phased removal of OPV [15]. In addition, an established surveillance for VDPV excretion in diagnosed PIDs will help to maintain the eradication status. 

As of May 2020, only 149 cases of iVDPV have been reported worldwide between July 2018–Jan 2020 [13]. Most of the reported cases that were detected through AFP surveillance had paralysis before they receive a diagnosis of immune deficiency. China reported eleven cases of iVDPV using AFP surveillance throughout 2021 [7]. Other iVDPV cases have been detected during the detection of enterovirus infection in children with suspected or confirmed PID through stool cultures. Among the iVDPV cases reported, 56% of iVDPVs were type 2, 23% were type 3, 17% were type 1, and 4% were heterotypic mixtures [6]. However, after the global removal of OPV2 from routine immunization in 2016, the incidence of iVDPV2 detection declined markedly. In India a total number of 12 iVDPV cases had been reported during the years 2009–2016 (ERC data, unpublished) from which 11 cases had been reported by the AFP surveillance and one case from a single-center PID pilot study initiated by us in 2014 [9]. We have expanded the single-center study by involving a total number of seven FPID-recognized PID diagnostic centers in India, in collaboration with WHO.

In this study with 154 PID patients enrolled at seven study sites across India, we found that these patients are highly susceptible to enteroviruses with a percentage positive of 21.42 in which PV positivity is 2.59% and NPEV positivity is 18.83%. Our previous study with a single site reported a similar result of 19% EV positivity in PID cases; however, the PV positivity was 7% and NPEV was 12% [9]. The difference in PV and NPEV positivity observed between both the studies is mostly due to the multicentric approach in comparison to a single-center enrolment, revealing higher NPEV positivity. Estimation of the prevalence of poliovirus excreters among PID patients of 13 OPV-using countries reported a very low prevalence (5.19%) of enteroviruses in those countries as compared to the Indian patients observed in our study [16]. Out of 154 cases studied, 14 were lost to follow-up during the study period due to varied reasons. Among them, seven cases were lost at the third sample which was to be collected after 6 months from the second collection; and three patients were lost at the second sample collection.

Patients with combined immunodeficiency have been reported to possess 10-folds higher risk of excreting poliovirus compared to patients with predominantly antibody defects, as per the data of the poliovirus prevalence study of the 13 OPV-using countries [16]. Similarly, the patients with combined immunodeficiencies had an increased risk of prolonged poliovirus excretion compared to primary antibody deficiencies as observed in our study. In our study, iVDPV1 was detected in a patient with combined immunodeficiency (P1) two of the patients who had <1% divergence in the poliovirus genome expired before more follow-up samples could be collected (P2, P3). One XLA patient who shed P3SL did not show any mutations (P4). The 3-year-old hyper IgM patient (P1) who was positive for iVDPV1 had a confirmed CD40 mutation using Sanger sequencing. Patients with a CD40L or CD40 deficiency have markedly reduced IgG levels, low IgA levels, and normal to elevated IgM levels. Clinically, CD40 deficiency is characterized by defects in cellular and humoral immunity resulting in a susceptibility to recurrent sinopulmonary bacterial infections and severe opportunistic infections. Interestingly, the iVDPV1 patient in our study with Hyper IgM syndrome had sufficient antibody level to PV1 pre- and post-iVDPV1 detection. Furthermore, the IL-8 secretion during iVDPV clearance may be associated with the activation of macrophages, neutrophils, or monocytes in absence of B cell and T cell interactions (Appendix A) [17]. The observation indicates that, beyond B cell defects, patients’ susceptibility to opportunistic pathogens might be due to impaired T cell and innate immune responses [18]. Hyper IgM syndrome was originally considered as humoral primary immunodeficiency but is now considered as combined immunodeficiency because of the clinical features and the defect of T-cell priming, resulting from a defective T–B cell or dendritic cell interaction [19]. It could be the reason for four enterovirus positivity out of nine hyper IgM cases enrolled in our study (Appendix A). It would be interesting to investigate what makes this category of patients more susceptible to EV than other combined immunodeficiency disorders. 

iVDPV was detected in P1 about 12 months from the last OPV dose, whereas the other patients in which no or <1% divergence from the genome was observed had received their last dose of OPV 1–5 months before detection. This reiterates that the early diagnosis of a PID patient and prompt initiation of VDPV surveillance soon after diagnosis of these patients can help in timely detection and help in halting transmission.

Non-polio enteroviruses may be associated with polio-like paralysis after the eradication of polioviruses and may also result in aseptic meningitis, encephalitis, paralysis, myocarditis, and neonatal enteroviral sepsis [20], resulting in outbreaks [21]. As such, surveillance for NPEVs in addition to polioviruses especially as a part of environmental surveillance [22] and in high-risk groups is warranted and is routinely performed in many countries [23,24,25]. NPEV excretion was detected in 18.5% of cases in our study which is higher than our single-center study (12%) reported 5 years back [9], EV-B species being the commonest (79% isolates) followed by EV-A (19% isolates). Further, our current study with multiple sites has identified long-term NPEV excreters which was not observed in the single-center study. Our data on the changing pattern of NPEVs in India with an increase in the NPEV percentage and number of long-term excreters among PID patients highlight the need to set up effective NPEV surveillance program owing to their involvement in meningitis, acute flaccid myelitis, encephalitis, and other severe neurological conditions worldwide [26]. In a Tunisian study the NPEV detection rate was reported to be 12.4% among PID patients and the predominant species was EV-B, with the circulation of viruses from species EV-A [27]. The relative abundance of NPEVs may favor recombination events between PVs and NPEVs in co-infected children, thus leading to the recurrent emergence of recombinant VDPVs as seen in some region of Madagascar [28]. Long-term excretion of E16 and E21 (EV-B species) in adult CVID patients as observed in our study is in a similar line to an Israel study wherein echovirus 11 (EV11) strains were isolated from a chronically infected adult patient with CVID for a period of 4 years. The establishment of chronic NPEV infections in these patients requires the adaptation of the viruses in the host while maintaining viral counter defense mechanisms [29]. 

The majority of our cases detected positive for NPEVs and all the patients who tested positive for poliovirus serotypes received IVIg during the course of treatment. Although the IVIg preparations contained sufficient titers to poliovirus [30], poliovirus excretion was observed, possibly due to irregular treatment, variable–specific antibody kinetics, and lack of T-cell-mediated immune response in PID patients. Furthermore, protection at mucosal sites is uncertain which is the main portal of entry for enteroviruses [31]. In a similar report, IVIg failed to reduce the incidence of hospital-acquired infections in the infants in a study conducted by The National Institutes of Health-sponsored Neonatal Research Network [32]. No difference in enterovirus infection was observed between the IVIg receivers and non-receivers in our study.

Major limitations of our study include the restriction of the study population to a few Indian states, and irregular sample collection which happened during the COVID-19 lockdown. Another limitation of the study was the absence of a control group (healthy subjects with the same age range) to compare the rate of viral shedding with PID patients. To address the limitations of the study population, the study has been expanded to include more centers in different parts of the country. The sample collection for the cases is being carried out through the already-established NPSP network which is a very robust system for timely collection and transport of the stool specimens to the laboratory. 

Our study with multiple sites across India and the involvement of NPSP, WHO, for the collection of samples during the COVID-19 pandemic has established a standard protocol for PID surveillance in India. The study demonstrated an increase in enterovirus prevalence compared to the previous data; although, the risk of chronic excretion of poliovirus among PID patients in India was found to be low. The successful completion of the Phase-1 study paved the way for further expansion (phase-II) including more referral hospitals across all parts of India and integrated PID surveillance with national AFP surveillance for the continuous screening of PID patients until complete OPV cessation. In addition, the study also suggests monitoring NPEV prevalence in order to avoid recombination and to develop effective prevention and control strategies [33]. In conclusion, additional and timely surveillance for VDPVs in PID patients is a robust strategy to maintain a polio-free status in India.

## Figures and Tables

**Figure 1 vaccines-11-01211-f001:**
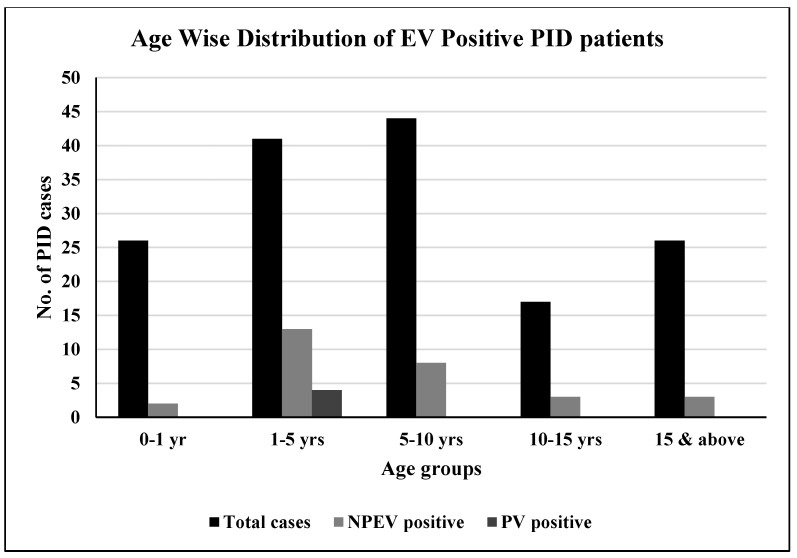
Age wise distribution of EV-positive PID patients enrolled in the study: The PID patients enrolled in the study during Dec 2019 to Dec 2021 were divided in to groups as per EV excretion. EV—Enteroviruses, NPEV—Non-polio enteroviruses, PV—Polioviruses, PID—Primary immunodeficiency disorders.

**Figure 2 vaccines-11-01211-f002:**
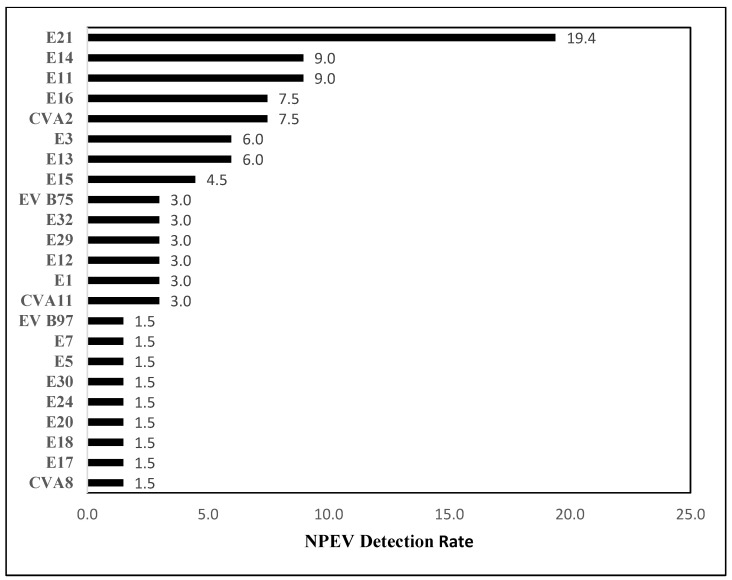
Detection rate of NPEVs among PID patients enrolled in the study. EV—Enteroviruses, NPEV—Non-polio enteroviruses, E—Echoviruses, CVA—Coxsakievirus A.

**Figure 3 vaccines-11-01211-f003:**
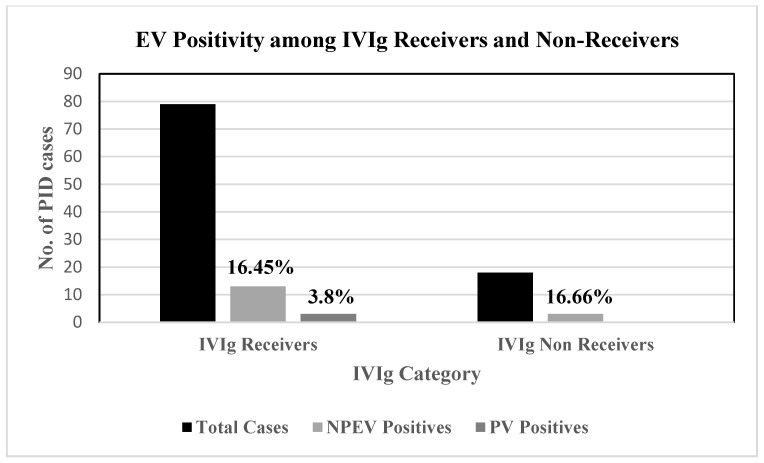
Enterovirus positivity among IVIg receiving and non-receiving PID patients enrolled in the study. EV—Enteroviruses, IVIg—Intravenous Immunoglobulin, NPEV—Non-polio enteroviruses, PV—Polioviruses.

**Figure 4 vaccines-11-01211-f004:**
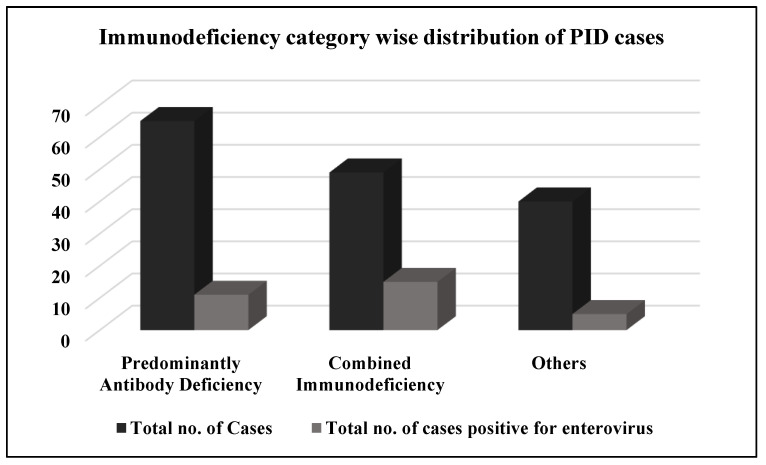
Immunodeficiency category wise distribution of PID cases enrolled in the study.

**Table 1 vaccines-11-01211-t001:** Enrolment of cases and lab investigations.

Total PID cases enrolled for the study	157
Total PID cases enrolled and sample collection done	154
Total cases enrolled but deceased before sample collection	03
Total number of stool samples received	535
Total deceased cases during the study period	17
Total poliovirus (PV) positive cases	04
Total non-poliovirus enterovirus (NPEV) positive cases	29

**Table 2 vaccines-11-01211-t002:** Detection of polio and non-polio enteroviruses from study sites * (December 2019–December 2021).

Study Sites	Total Cases Tested	Total Samples Received	Enteroviruses Detected			
			PV Positive Cases	Types of PV Cases	NPEV Positive Cases	Types of NPEV Cases
B.J.W.HC, Mumbai	71	244	03	iVDPV1, P3SL, P1SL	14	E30, E5, E14, E11, E20, EVB97, CVA11, E12, EVB75, E13, E3, E32, E7, E29
ICMR-NIIH, Mumbai	21	67	00	-	05	E18, E13 (2), CVA2, E21
SGPGIMS, Lucknow	14	63	00	-	04	E16, CVA2, EVB75, E21
GMC, Kozhikode	14	52	00	-	01	E3
ASTER CMI, Bangalore	13	41	00	-	01	CVA2
KMC, Mangalore	12	36	01	P3SL	01	CVA2
NIMS, Hyderabad	09	32	00	-	03	CVA8, E17, E29
Total	154	535	04	-	29	-

iVDPV, Immunodeficiency-associated vaccine-derived polioviruses; P3SL, Sabin-like poliovirus type 3; P1SL—Sabin-like poliovirus type 1; E30, Echovirus30; E5, Echovirus5; E14, Echovirus14; E11, Echovirus11; E20, Echovirus20; EVB97, EnterovirusB97; CVA11, CoxsackievirusA11; E12, Echovirus12; EVB75, EnterovirusB75; E13, Echovirus13; E3, Echovirus3; E32, Echovirus32; E7, Echovirus7; E29, Echovirus29; E18, Echovirus 18; CVA2, CoxsackievirusA2; E21, Echovirus21; E16, Echovirus16; CVA8, CoxsackievirusA8; E17, Echovirus17. * Sample collection dropped down during April 2020 to June 2020 due to COVID-19 pandemic.

**Table 3 vaccines-11-01211-t003:** Demographic and sequencing details of PID patients detected positive for PV infection.

PID Type	Age (Years)/Gender	IVIg Therapy	Months from Last OPV	Results by Sample Collection Day (D)	Isolated PV	No. of Mutations	Synonymous Mutation	Non-Synonymous Mutation	Amino Acid Changes	Final Status
Hyper IgM syndrome	3 yrs/Male	Yes	12	D01: iVDPV1; D80: Neg; D110: Neg	iVDPV1	16	G126A	A296C	Lys99Thr	Neg
							T207C	A316T	Thr106Ser	
							A213T	A503G	Glu168Gly	
							C243T	A641G	Lys214Arg	
							T345C	G892A	Asp298Asn	
							C450T			
							C558T			
							C711T			
							A732T			
							A849T			
							G870A			
SCID	1 mth/Female	No	1	D01: P1SL	P1SL	4	C258T	A299G	Asn100Ser	Deceased before next sample collection
							T364C			
							A903G			
WAS	5 mths/Male	Yes	5	D01: P3SL; D34: Neg; D104: Neg	P3SL	3	G99A	C161T	Ala54Val	Deceased
							G498A			
XLA	4 yrs/Male	Yes	1	D01: P1SL + P3SL; D29: P3SL; D59: Neg; D91: Neg	P3SL	0	NA	NA	NA	Neg

SCID, Severe combined immunodeficiency; WAS, Wiskott Aldrich Syndrome; XLA, X-linked Agammaglobulinemia; yrs, age in years; mths, age in months; Neg, Negative for enteroviruses.

**Table 4 vaccines-11-01211-t004:** Details of PID patients detected positive for NPEV infection.

Sr. No.	PID Type	IVIg Prophylaxis	Result	Species	NPEV Excretion (In Days)	No. of Positive Samples	Final Status
1	CVID	Yes	E16	B	304	5	Negative
2	CVID	Yes	EVB75	B	-	1	Negative
3	CVID	Yes	E21	B	507	12	Positive
4	CVID	Yes	E13, E14	B	-	1, 1	Positive
5	CVID	Yes	E13, E14	B	-	1, 1	Positive
6	CVID	Yes	CVA2	A	-	1	Negative
7	WAS	No	E20	B	-	1	Deceased
8	WAS	Yes	E11, E18	B	-	1, 1	Negative
9	WAS	Yes	E11, EVB97	B	-	2, 1	Negative
10	WAS	Yes	E14, E3, CVA11	C	80	1, 1, 2	Positive
11	Hyper IgM syndrome	Yes	CVA2	A	-	1	Negative
12	Hyper IgM Syndrome Type 2 + Hypogammaglobulinemia	Yes	E5	B	-	1	Negative
13	Hyper IgM syndrome	Yes	EVB75	B	-	1	Negative
14	Ataxia Telangiectasia with IgA deficiency	Yes	CVA8	A	-	1	Deceased
15	Ataxia Telangiectasia with IgA deficiency	Yes	E17	B	-	1	Positive
16	Ataxia Telangiectasia	No	E15, E29	B	-	3, 1	Negative
17	Ataxia pancytopenia	No	E29	B	-	1	Negative
18	TRNT1 Deficiency	Yes	E1, CVA2	A	77	2, 1	Negative
19	TRNT1 Deficiency	Yes	E24, E32	B	93	1, 2	Negative
20	Auto inflammatory disease	Yes	E12	B	65	2	Negative
21	CHS + EBV Viremia	No	E30	B	-	1	Negative
22	CMC-STAT1-GOF	No	E14	B	55	3	Negative
23	DOCK-8 Deficiency	No	E11	B	91	3	Positive
24	Hypogammaglobulinemia	No	E21	B	-	1	Negative
25	MSMD	No	CVA2, E13, E3	B	47	1, 1, 2	Negative
26	SCID	Yes	E3	B	-	1	Negative
27	SPENCDI	No	E13	B	-	1	Negative
28	XLA	Yes	CVA2	A	-	1	Negative
29	ZBTB24	Yes	E7	B	-	1	Negative

CVID, Common variable immunodeficiency; TRNT1, CCA-adding transfer RNA nucleotidyl transferase enzyme; CHS, Chediak Higashi Syndrome; EBV, Epstein–Barr virus; CMC-STAT1-GOF, Chronic mucocutaneous candidiasis–Signal transducer and activator of transcription 1–Gain-of-function; DOCK8, Dedicator of cytokinesis 8; MSMD, Mendelian susceptibility to mycobacterial diseases; SPENCDI, Spondyloenchondrodysplasia with immune dysregulation; ZBTB24, Zinc Finger and BTB Domain Containing 24.

**Table 5 vaccines-11-01211-t005:** Status of patients detected positive for enteroviruses.

Patient Category	Total EV Positive	PV Positive	NPEV Positive
Alive and excreting virus	4	-	4
Alive and stopped excretion of virus	20	2	18
Patient deceased	4	2	2
Patient lost to follow-up	5	-	5
Total	33	4	29

## Data Availability

The raw data supporting this article will be made available by the authors, without undue reservation.

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
