# Peer review of "Assessment of Enterovirus Excretion and Identification of VDPVs in Patients with Primary Immunodeficiency in India: Outcome of ICMR–WHO Collaborative Study Phase-I"

_vaccines, 2023, doi:10.3390/vaccines11071211_

Round 1
Reviewer 1 Report
The authors present a study carried out in different regions of India related to the excretion of enterovirus and vaccine-derived poliovirus (VDPV), in a group of patients with primary immunodeficiency.
In a sample of 154 patients, the prevalence of enterovirus and VDPV was 21.42 and 2.59%, respectively.
Overall, the authors conclude that children with primary immunodeficiency are a potential source of VDPV shedding and that, in some cases, these patients can shed the virus up to one year after receiving oral polio vaccine. In addition, detection of non-polio enteroviruses were the main detected viruses excreted by patients.
In my opinion the article is interesting, however, I consider that the information is more important at the local level, due to the advances in the design of vaccines against polio. Likewise, the spread of non-polio enteroviruses in patients with primary immunodeficiency has no comparison group.
I know that at this point it is probably impossible to compare the rate of shedding of the virus in a control group (healthy children with the same age range), however in my opinion this is an important point in the desing of the study
Furthermore, it seems important to include the table of cytokine results in the different groups.
Author Response
Point 1: In my opinion the article is interesting, however, I consider that the information is more important at the local level, due to the advances in the design of vaccines against polio.
Response 1: We thank the reviwer for the very pertinent opinion. Yes, we agree that there are advances in polio vaccine design and application but OPV remains the major vaccine for global eradication program because it is cheap, easy to use and confers robust gut immunity. Although developed/rich cuntries are using IPV, recent polio outbreaks in New York state, London, and greater Jerusalem showed the impact of virus transmission within the countries jeopardising the entire program.
Outbreaks of circulating vaccine-derived poliovirus type 2 (cVDPV2) have become a global concern, as the world has seen more annual cases of cVDPV2 than wild poliovirus type 1 (WPV1) since 2017. While 27 outbreaks have been declared closed by the GPEI in the past two years, the scale and speed of disease spread pose a risk for global polio eradication. Children missing polio vaccinations creates opportunities for polio to re-emerge and spread – as seen in 2022 when WPV1 originating in Pakistan was detected in paralyzed children in Malawi and Mozambique. This episode served as a reminder that as long as polio exists anywhere in the world, it remains a threat to people everywhere. In November 2020, the novel oral polio vaccine type 2 (nOPV2) was authorized under the Emergency Use Listing (EUL) pathway by the World Health Organization (WHO) based on the studies for safety, reactogenicity, immunogenicity and the desired genetic stability. But unfortunately, just a few days back, the GPEI reported that seven children, six in the Democratic Republic of the Congo (DRC) and one in neighboring Burundi, had recently been paralyzed by poliovirus strains derived from a vaccine meant to prevent the disease. These are the first cases linked to a new polio vaccine that was painstakingly designed to avoid just this problem. Therefore, regardless of which polio vaccine is used to stop an outbreak, there must be high immunization coverage for all children to be protected against paralysis.
Point 2: Likewise, the spread of non-polio enteroviruses in patients with primary immunodeficiency has no comparison group.
I know that at this point it is probably impossible to compare the rate of shedding of the virus in a control group (healthy children with the same age range), however in my opinion this is an important point in the desing of the study
Response 2: We are very much thankful to the reviewer for pointing this out. We agree that inclusion of Non-PID controls in the study would have improved the study design. We would like to inform that as per the recommendation of the Polio Research Committee of WHO-HQ, the surviellance part of the study was aligned to match the WHO, global iVDPV surviellance protocol being followed in other countries. The idea was to integrate the iVDPV surviellance with AFP surviellance in future so that all PID cases can be screened. However we would definitely plan to incorporate Non-PID control in our further studies with due ethics approval.
Point 3: Furthermore, it seems important to include the table of cytokine results in the different groups.
Response 3: The study design involved the cytokine estimation of the iVDPV positive cases only. The table of cytokine results of the iVDPV case has been included as supplementary table in the revised manuscript.
Reviewer 2 Report
This paper describes a comprehensive study on EV excretion of PID patients in India. Provided information would be useful for the design of a PID surveillance and to evaluate the potential threat of PV excretion from PID patients in polio-free countries.
Specific comments:
- I might not understand whether the current study was conducted as a national surveillance based on the law or just a research study. This point should be clarified in the text (Introduction or Results) and in the Materials and Methods section.
- The current study involved seven hospital/study sites. Is it possible to estimate the whole number of PID patients in India?
- 3.4. Breakdown of CIV categories might be shown in Fig.4 or supplementary data as a new graph.
- Discussion: Patients with combined…. In this paragraph, An iVDPV case in this study was discussed; however, only one case was identified, so the relative importance of CID might not be clearly discussed.
- Discussion: This reiterates that …. This sentence should be clarified.
- Discussion: The establishment of chronic NPEV …. This sentence should be clarified.
- Discussion: The meaning of ‘Phase 1’ and ‘Phase II’ is not clear. Is this study a part of a clinical trial?
- Data availability: Useful law data may be posted as a Supplemental data set.
Author Response
This paper describes a comprehensive study on EV excretion of PID patients in India. Provided information would be useful for the design of a PID surveillance and to evaluate the potential threat of PV excretion from PID patients in polio-free countries.
We thank the reviewers for highlighting the importance of the study and emphasizing on its usefulness at current scenario.
Specific comments:
Point 1: I might not understand whether the current study was conducted as a national surveillance based on the law or just a research study. This point should be clarified in the text (Introduction or Results) and in the Materials and Methods section.
Response 1: WeThank the reviwer for pointing this out. The study was a research study proposed to WHO and approved by the Polio Research Committee, WHO-HQ, Geneva. The suviellance part of the study was aligned to Global iVDPV surviellance protocol as per the reccomendation of WHO. We have revised the sentences in the Introduction and Material and Methods of the revised manuscript to clarify this point (page 2 para 3 and 4)
Point 2: The current study involved seven hospital/study sites. Is it possible to estimate the whole number of PID patients in India?
Response 2: No. There are not many dagnostic centers and hospitals in India to diagnose PID. Although our Phase II study includes 20 hospitals across India, it is difficult to estimate the whole number unless the newborn screening facility is available in India.
Point 3: Breakdown of CIV categories might be shown in Fig.4 or supplementary data as a new graph.
Response 3: We thank the reviwer for the suggestion. The beakdown of CID and also PAD categories have been included as supplementary figures in the revised manuscript.
Point 4: Discussion: Patients with combined…. In this paragraph, An iVDPV case in this study was discussed; however, only one case was identified, so the relative importance of CID might not be clearly discussed.
Response 4: We thank the reviwer for pointing this. We would like to explain here that from the total 4 polio positive cases detected in the study, 3 were cases of Combined Immunodeficiency where in the virus has already started showing mutation as shown by nucleotide changes (Table-3). There was a single case of Predominantly Antibody Deficiency (PAD, XLA patient) with no nucleotie change. The data indicated the involvement both antibody and cell mediated immunodeficiency (CID) for prolong excretion of polioviruses leading to nucleotide changes.
Point 5: Discussion: The establishment of chronic NPEV …. This sentence should be clarified.
Response 5: Sorry for the inconvinience. The sentence has been modified in the revised manuscript (page 11, 3rd Para)
Point 6: Discussion: The meaning of ‘Phase 1’ and ‘Phase II’ is not clear. Is this study a part of a clinical trial?
Response 6: No, the study was not a paert of clinical trial. The research study was initially planned for 2 years. Based the significant outcomes of the study the Polio Research Committee. WHO-HQ extended the study for another 2 years as Phase II. So the first two years of the study was designated as Phase I and the extentsion was designated as Phase II.
Point 7: Data availability: Useful law data may be posted as a Supplemental data set.
Response 7: Thank you for the suggestion. Important raw data (2 tables and 1 figure) have been incorporated in the revised manuscrpt as supplementary data set.